# An Emotional Respiration Speech Dataset

Rozemarijn Roes
r.h.roes@students.uu.nl
Utrecht University

Francisca Pessanha
f.pessanha@uu.nl
Utrecht University

Almila Akdag Salah
a.a.akdag@uu.nl
Utrecht University

## ABSTRACT

Natural interaction with human-like embodied agents, such as so-cial robots or virtual agents, relies on the generation of realistic non-verbal behaviours, including body language, gaze and facial expressions. Humans can read and interpret somatic social signals, such as blushing or changes in the respiration rate and depth, as part of such non-verbal behaviours. Studies show that realistic breathing changes in an agent improve the communication of emotional cues, but there are scarcely any databases for affect analysis with breathing ground truth to learn how affect and breathing correlate. Emotional speech databases typically contain utterances coloured by emotional intonation, instead of natural conversation, and lack breathing annotations. In this paper, we introduce the Emotional Speech Respiration Dataset, collected from 20 subjects in a sponta-neous speech setting where emotions are elicited via music. Four emotion classes (happy, sad, annoying, calm) are elicited, with 20 minutes of data per participant. The breathing ground truth is col-lected with piezoelectric respiration sensors, and affective labels are collected via self-reported valence and arousal levels. Along with these, we extract and share visual features of the participants (such as facial keypoints, action units, gaze directions), transcriptions of the speech instances, and paralinguistic features. Our analysis shows that the music induced emotions show significant changes in the levels of valence for all four emotions, compared to the baseline. Furthermore, the breathing patterns change with happy music sig-nificantly, but the changes in other elicitors are less prominent. We believe this resource can be used with different embodied agents to signal affect via simulated breathing.

## KEYWORDS

datasets, emotions, respiration, embodied agents, social agents, emotion elicitation

**ACM Reference Format:**
Rozemarijn Roes, Francisca Pessanha, and Almila Akdag Salah. 2022. An Emotional Respiration Speech Dataset. In *INTERNATIONAL CONFERENCE ON MULTIMODAL INTERACTION (ICMI '22 Companion), November 7–11, 2022, Bengaluru, India.* ACM, New York, NY, USA, 10 pages. https://doi.org/10.1145/3536220.3558803

## 1 INTRODUCTION

One of the main challenges in modelling believable, human-like embodied agents is to bestow them with accurate non-verbal be-haviour. Non-verbal communication that uses body language, facial expressions, and paralinguistic cues constitute an important part of information that is relayed, and humans unconsciously decipher such cues. Among these cues, somatic reactions, such as skin re-sponses like a flush in the face, or a change in the breathing patterns also signal emotional changes. Breathing is an interesting physio-logical function, as normally it is regulated automatically, but can also be consciously controlled easily. Studies show a correlation between respiratory feedback and emotions such as joy, anger, fear and sadness [5, 13, 30]. When transferred to VR environments, we see that adding respiratory cues to virtual agents help humans de-tect the displayed emotions more accurately [10]. In an interesting study, Terzioğlu et al. have shown that adding breathing motions to a robotic arm tasked with carrying out manipulation tasks along-side human coworkers improves its appeal and life-likeness of the robot [41].

To enhance emotion embodiment with the help of respiration, we need to map the relation between different breathing patterns and distinct emotions. This can be achieved by collecting breath-ing ground truth data under elicited or naturally occurring emo-tional states. Preparing such a resource requires addressing sev-eral methodological and observational challenges. First of all, even though the link between emotion and respiration is well established in the psychology literature, we see reports showing that different emotions can trigger similar respiratory reactions. We need more observations dedicated to respiration to uncover stronger links. Secondly, tools for emotion measurements are not reliable: ques-tionnaires that capture self-reported experiences show that people cannot capture/report their emotions with 100% clarity. Measure-ments based on human annotations, i.e. second hand reports, also do not have a high agreement. Using somatic aspects of emotions offer a third alternative for this purpose [38]. Lastly, most of the emotional datasets are created with a passive emotional elicitation method, where participants are given images or videos with emo-tional content. These setups rarely have a speech component, i.e. the participants are mostly silent during the experiment [14, 39]. On the other hand, there are several emotional speech datasets where the participants are asked to mimic an emotion while per-forming an utterance [45]. Neither of these types of emotional datasets replicate the conversational or spontaneous speech, and lack in ecological validity. While breathing has been investigated in non-verbal interaction dynamics, such as turn taking [33], there is a lack of emotional speech datasets with respiration ground truth.

To address these challenges, this paper explores the connection between elicited emotions and changes in respiration by providing a new dataset that contains emotional respiration ground truth during speech. Instead of visual stimuli, we use music as our main

emotion elicitation method. In addition to the breathing patterns during spontaneous speech and listening episodes, we also provide self-reported emotion labels. The introduced Emotional Respiration Speech (ERS) dataset contains 400 minutes of recording, with four elicited emotions (i.e. happiness, sadness, annoyance, and calmness) that can provide useful breathing behaviour in a wide range of task settings.

This paper is structured as follows. In Section 2, we proceed with a discussion of related work on the relation between respiration and emotions, as well as the use of respiration in embodied agents. Section 3 introduces our experiment design for emotion elicitation. In Section 4, we present our analysis of the dataset focusing on the link between breathing patterns and emotions and conclude with a discussion in Section 5.

## 2 RELATED WORK

### 2.1 Human respiration and emotions

Breathing is part of the autonomic nervous system (ANS) which regulates involuntary physiological processes including heart rate, blood pressure, skin temperature/blood flow/sweating, gut motility, pupil size, and piloerection (i.e. bristling of hairs) [18]. The link between respiration, emotion and cognition is observed in experimental setups in numerous studies, and the underlying mechanism is further explored in neurobiology [2, 22, 43]. In this subsection we focus on the former, summarising observations between breathing patterns and specific emotions.

The main function of breathing is gas intake, which happens automatically. However, the ANS is also responsive to the environment, and to emotional demands. The debate whether specific emotions have distinct physiological reactions continues, with strong support for the idea [29]. Changes in the breathing rhythm for basic emotions is documented in various emotion elicitation experiments, however a consistent rhythm fingerprint is hard to assign to each emotion. For example, Santibañez and Bloch [35] reported that with fearful and anxious events, the participants showed significantly shallower respiration. Anger increased the respiration rate (RR), but caused no changes in depth. An earlier study documented the increase in RR, but not in depth [3]. A challenge here is that opposite emotions can trigger similar reactions. For example, sad participants, while actively crying, and happy participants, while laughing, both showed an increase in breathing depth [35].

Disgust and anticipatory pleasure both slow the breathing, but due to different reasons. In disgust, it is speculated that to expel the foul smells people tend to take less breaths. Another challenge comes from how the same emotion can have various formats. For example, Kreibig's survey of 134 articles on the literature reports that anger increases the respiration rate but seems to cause a variable depth [18]. A reasonable explanation in the variation of depth might have to do with the fact that anger can be covert or overt.

We can still take some general conclusions from the literature, despite the variability in results. A rise in arousal during negative emotions (such as fear, anger and anxiety) creates shallower and rapid breathing [5, 21]. Similarly, positive emotions with high arousal increase the respiration rate, whereas low arousal decreases it [18]. In short, the levels of arousal, as well as valence, seem to trigger similar variations in breathing rhythm, and similar influences of

different emotions on breathing can be explained via commonalities between them [4]. Figure 1 provides an overview of the effects of emotions on emotional dimensions (valence and arousal) and on respiration features, based on existing literature.

### 2.2 Respiration and emotion regulation

Breathing can be regulated at will, and controlling breathing for relaxation or reducing stress has a long tradition in various cultures and practices, such as doing yoga or Tai Chi [19, 25]. The question of how (and if) emotions can be regulated with such controlling techniques has been the focus of many studies. In this section, we briefly outline the prominent findings from these studies, as they can be useful to the effort of increasing the believability of virtual agents via breathing cues.

Detrimental effects of stress, negative emotions, and sympathetic dominance of the autonomic nervous system can be counteracted by different forms of meditation, relaxation, and breathing techniques [7, 15]. An early study on hatha-yoga practitioners demonstrated how the control of posture and manipulation of breathing significantly lowers the respiration rate in comparison to a control group with the same characteristics [40]. A subsequent study from the same laboratory provided evidence to support claims by yogi masters that a specific form of slowed respiration, i.e. rapid inhalation followed by slow exhalation at a reduced respiratory rate, was an effective technique for reducing physiological arousal when anticipating and confronting a threat [8]. In sum, voluntary slowing of respiratory rate under stressful conditions reduces physiological arousal as measured by skin resistance, finger pulse volume, and self-reports of anxiety [24]. Slow, deep breathing affects vagal modulation of the heart and even the functioning of certain neural circuits of the prefrontal cortex and amygdala, which are involved in emotional regulation [12].

In the medical literature, we find similar observations. A breathing regulation technique called Lamaze breathing is commonly used for childbirth, helping with relaxation and decreasing pain perception [44]. Breathing control during panic attacks to modulate emotion is a commonly suggested method [9, 16], as slower and deeper breathing patterns are observed to create a change in emotions towards positive [23]. Respiratory control is applied for long term treatments such as improving mood, enhancing learning [11] or even changing smoking behaviour [20]. An interesting study where the participants were taught specific breathing patterns reports changes in emotional states, even though the participants were not aware of the process, showing that the regulation of breathing in specific ways will affect emotions regardless of the state or expectations [31].

### 2.3 Respiration in Embodied Agents

The relationship between emotion and respiration can be used to the advantage of virtual agents, either to evaluate the user's affective state in human-computer interfaces or to enhance the emotion synthesis and perception of the virtual agent.

Modelling realistic emotionality is a big challenge in producing virtual agents. In the case of speech, the emotionality will depend not only on the linguistic and voice characteristics but also on non-verbal cues such as laughter, deep breaths, or sighs. Due to

| Emotion | Emotional dimension | | Respiration feature | | |
|---|---|---|---|---|---|
| | Valence | Arousal | Respiration rate | Respiration depth | Respiration regularity |
| **Anger** | Low | High | Increase | Increase | Potentially irregular |
| **Fear** | Low | High | Increase | Increase | Potentially irregular |
| **Anxiety** | Low | Medium to high | Increase | Decrease | *Unclear* |
| **Crying sadness** | Low | Medium to high | Increase | Variable | Irregular |
| **Non-crying sadness** | Low | Low to medium | Increase | Variable | *Unclear* |
| **Calmness** | Medium to high | Low | Decrease | Decrease | *Unclear* |
| **Happiness** | High | Medium to high | Increase | Variable | Irregular (when laughing) |

**Figure 1: An overview of the effects of emotions on emotional dimensions (valence and arousal) and on respiration features, based on the existing literature.**

the role of breathing in sound production, a better understanding of breathing patterns during emotional moments or respiratory events would provide helpful information for more realistic interactions. In this context, Niewiadomski et al. studied the breathing phases during laughter and how they influenced the perception of laughter intensity and used breathing in a multimodal modality for believable laughter synthesis [26, 27]. More importantly, changes in respiration patterns of agents change agents' perceived arousal and emotionality. Klausen et. al report that even in the absence of other expressions, with a simple soft robot simulating only respiratory movement, emotions are detected correctly [17]. In a related study, Terzioğlu et al. added simple, rhythmic breathing motions to robotic arm manipulators on a factory ground, which improved the perception of the robots by the human co-workers [41]. These behaviours are examples of *Secondary Actions*, and while they do not serve a specific communicative purpose, they increase the life-likeness of the agent [32, 42].

The relation between breathing and emotion has also been studied in multimodal interactions using humanoid agents. De Melo et al. developed virtual humans as a model to simulate respiration by animating respiration videographically [10]. Fourteen emotions and their accompanying respiration patterns were chosen: excitement, relaxation, focus, pain, relief, boredom, anger, fear, panic, disgust, surprise, startle, sadness and joy, respectively. They used several respiration parameters including respiration rate, respiration depth, and respiration curve, which defines how the depth differs over the respiration cycle. The researchers showed videos of the virtual humans expressing emotions with or without these respiration patterns. For example, a virtual human showing sadness, would display a sad facial expression and neutral posture without visible respiration in the control condition. In the respiration condition, the agent had a slow and deep respiration. The participants had to state which emotion they were perceiving in the agents. The results of the study showed that it was easier to detect emotion when the agents were displaying excitement, pain, relief, anger,

fear, panic, boredom and startle. There was no significant difference for relaxation, disgust, surprise, sadness and joy and even a negative effect regarding perceiving focus correctly. Even though these results were not significant for every emotion they tried to portray, they do make clear that humans use some sort of induction of one's emotions by observing the respiration.

Intuitively, the respiratory patterns of the user will also contain significant information about their emotional state, motivating their use as biofeedback for virtual agents to better adapt their interaction. This biofeedback is particularly relevant for virtual therapy and coaching, where controlling breathing is often used for emotion adaptation. Shamekhi et al. have explored this idea by monitoring the respiration of users using a breathing belt and utilising the measured breathing rate to evaluate the user's emotional state and whether they are responding to the directions of a meditation coach [36, 37]. Depending on the user's performance, the coach can then provide personalised feedback. The feedback helped create a more efficient meditation session compared to a videotaped meditation and made the interaction more organic. In this scenario, providing alternatives for automatically predicting the breathing signal and removing the need for breathing belts would improve the usability in real-world scenarios.

A last example is the chat agent Paola, which is designed with a respiration model to test if the addition of realistic breathing adds more rapport and natural perception of a virtual agent [28]. The expectation of the researchers were to see a greater rapport in the breathing condition, however the results suggested no significant difference in the perception of naturalness. However, the respiration model used by this study synthesised breathing instances during silent episodes, i.e. before or after speech. Moreover, the respiration was generated only visually and not in the audio channel, nor were there any emotional respiration patterns used. All of these shortcomings could have affected the impact of respiration perception. With the help of the database we collect in this paper,

it will be possible to investigate the effects of emotional breathing during speech in different agents.

## 3 EXPERIMENT DESIGN

The objective of the research reported in this paper is to record breathing patterns during speech and non-speech segments under induced emotional responses. The affect is expressed in the dimensions of valence and arousal. To examine to what extent they change certain respiratory features that may underlie these experiences, we use music-induced emotions. We designed an experiment in which two groups of participants had to listen to four different emotional types of self-selected music, including happy, sad, calm, and annoying pieces. The emotions we chose represent all four quadrants of the circumplex model by Russell [34]: happiness representing medium to high arousal and high valence; sadness low to medium arousal and low valence; calm representing low arousal and medium to high valence; and annoying representing high arousal and low valence, visualised in Figure 2. In this section, we further detail the experiment setup.

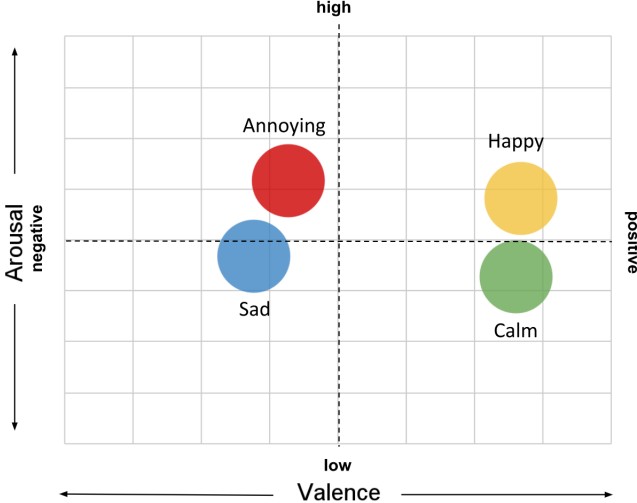

**Figure 2: The circumplex model, with the mean arousal and valence of the four emotions as reported in our experiment. The x-axis represents valence, where the most negative valence score is to the left side and the most positive score to the right. The y-axis represents arousal. The most active score is on the top of the graph and the most passive score on the bottom.**

### 3.1 Participants

Participants were recruited through social media via convenience sampling. In total, twenty people took part in the experiment. Of those, nine were men (45%), ten were women (50%) and one person indicated their gender as 'other' (5%). They mostly consisted of university students and their age ranged from between 18–21 years old (40%), to 22–25 years old (55%) and 26–29 years old (5%) (mode: 22–25 years).

### 3.2 Survey

Instead of using a music emotion elicitation database, we decided to let participants choose songs that are meaningful for them. Self-selected songs with which they had personal emotion connections may approximate real-life emotions better. Two to three weeks before the experiment, each participant filled in a digital survey to submit songs that made them either happy/sad or calm/annoyed. Participants were randomly and evenly divided into one of either emotion pairs, through a randomising option in the survey. For each emotion category, we collected at least three songs, where each song was weighted on a 1-5 Likert scale on how intensively a song triggers the emotion (e.g. 1: a little bit happy; 5: extremely happy). For each participant, we selected two songs with the highest intensity scores from each category. We prepared ninety second fragments for all the selected songs to be used during the experiment. At the beginning of the survey the participants filled out an informed consent form and an exclusion criteria form. The experimental protocol is vetted by the university ethics board for experimenting with human subjects.

### 3.3 Instruments

The affective states of the participants were measured through a digital version of the Self-Assessment Manikin (SAM) on a 9-point pictorial scale for valence and arousal values [6]. These dimensional variables were chosen to avoid self-report biases as participants may remember which songs they filled in for which emotion category. Respiration was measured with the biosignalsplux©piezoelectric respiration sensor (PZT: a respiration belt) and a Bluetooth controlled microphone attached to the neck. One PZT was placed around the thorax and one around the abdomen. Based on research by biosignalsplux©, we placed the thoracic sensor on the lateral side of the body two centimetres under the nipple line, which results in the highest peak to peak amplitude. For the same reason, we placed the abdominal sensor frontally, between the eighth and tenth ribs. The movement signals (measured in Volts, with a range of -1.5 to 1.5) were recorded with OpenSignals (r)evolution software [1]. Calibration measurements were done by instructing each participant to inhale and exhale as deep as possible for multiple times. The microphone receiver was connected to a Canon Legria HF650 video camera, so that the audio and video were synchronously recorded.

### 3.4 Experiment flow

The experiment consisted of one session for each participant and lasted approximately 30 minutes (Figure 3). Participants were first given a short explanation of the experiment flow, then the belts and the microphone were attached and a test round was recorded. At this point, the experimenter left the room to avoid observer effects. The baseline condition after calibration lasted 45 seconds. After that, four musical fragments were played in a semi-random order for each participant to cancel out order or fatigue effects. All the songs of one emotion condition were always played one after the other. After each song-fragment, participants first filled in the SAM questionnaire, and then talked about their experience of listening to the song for a maximum of 45 seconds (reflection). That was followed with an inter-stimulus period of 30 seconds to reduce possible emotional effects induced by the fragments that

were played before. An experiment interface on a monitor guided the participants through every step.

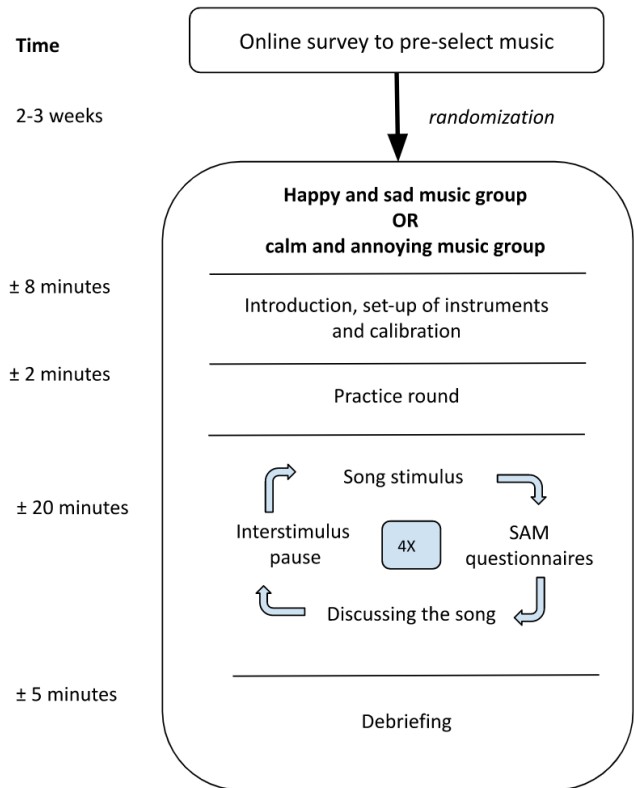

**Figure 3: Experiment flow. The segments of the experiment are given with time indices on the left.**

## 3.5 Database Features

The Emotional Respiration Speech Dataset has raw files in the format of video and audio files, as well as respiration belt signals. We converted these raw files into various formats to share with the research community. Figure 4 shows the input and output files of our database. Thus, we share SAM questionnaire results for each song, the transcriptions of speech episodes, the facial keypoints, gaze and head directions of each participant, as well as the paralinguistic features for speech and silence episodes.

## 4 RESULTS

### 4.1 Measurement Units

For self-reported valence and arousal levels we use the SAM questionnaire results recorded after each song. The participants used a 9 point pictorial Likert scale to document how much they experienced valence and arousal for the given emotion. For the breathing features, we chose respiration frequency (in breath cycles per minute) and peak to peak distance (in seconds) to represent respiration rate. For the depth, we calculated a proportional measure by adding the

peak and minimum amplitude values of a respiration cycle together (in volt).

### 4.2 Self-reported emotions

To evaluate whether the songs truly elicited expected emotions, we analysed the self-reported valence and arousal levels. The results showed that songs for each emotion type influenced the valence levels of emotion strongly, in the expected manner, as shown in Table 1. However, there were no significant effects on the experienced arousal levels. When looking at non-significant effects, the happy music did seem to increase arousal, as did annoying music. Sad music did not have any clear influence on arousal. Lastly, calm music seemed to decrease arousal, but only after the second song (for the separate song values, see Appendix, Table 3). These results indicate that it takes more time to influence arousal with music.

**Table 1: Mean valence and arousal values during music listening for the two condition pairs and their baseline. The asterisk (\*) indicates a significant difference in values between the emotion category on dimensional values and no music (baseline).**

| Music emotion | Mean valence (SD) | F (p-value) | Mean arousal (SD) | F (p-value) |
|---|---|---|---|---|
| No music (N=11) | 7.00 (±0.89) | | 4.73 (±1.42) | |
| Happy (N=11) | 7.68 (±0.87) | | 5.82 (±1.37) | |
| Sad (N=11) | **3.77 (±1.03)\*** | 45.7 (**<.001**) | 4.68 (±0.96) | 2.97 (.063) |
| No music (N=9) | 6.78 (±0.67) | | 4.89 (±1.05) | |
| Calm (N=9) | **7.61 (±0.74)\*** | | 4.28 (±2.20) | |
| Annoying (N=9) | **4.28 (±1.58)\*** | 28.18 (**<.001**) | 6.17 (±1.32) | 3.34 (.059) |

Even though not all results are significant, they were still close to the expected effects of music elicitation as reported in the literature. The lack of significant effects may be due to the small sample size and due to the need of longer emotion elicitation.

### 4.3 Respiration features

We will demonstrate here the measured respiration patterns by taking two participant's respiration as an example. Figure 5 displays forty five second fragments of the chest and abdomen respiration patterns during the baseline (i.e. no music), as well as during listening to songs and the speech episodes that come after listening to music. The first noticeable difference between the two participant is the baseline respiration, which demonstrates how much respiration patterns differs between individuals. The first participant who listened to sad and happy songs took deeper breaths than the second participant who listened to annoying and calm songs (see Figure 5, upper and lower parts of the figure for comparison) .

When we compare chest and abdominal respiration (see Figure 5) for both participants, we observe that the chest respiration to be more sensitive to the effects of different emotional songs compared to the abdominal respiration, as the abdominal signal has a similar number of respiration cycles and amplitude for each song. The music condition induces deeper breathing and also more amplitude variability, compared to the no-music condition, across the experiments.

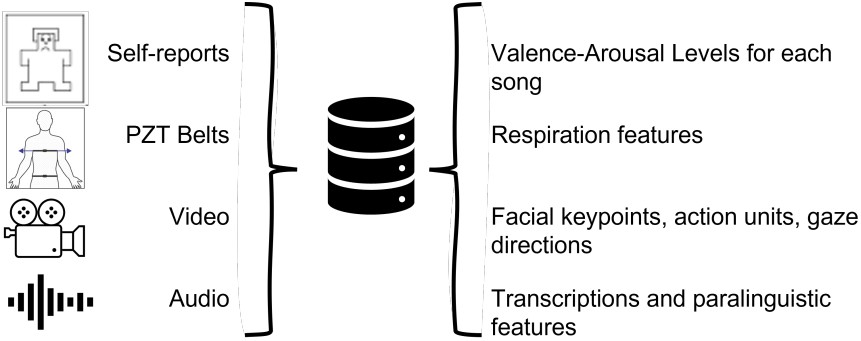

**Figure 4: The Emotional Respiration Speech Dataset**

A last observation can be made between listening and speaking: again, for both participants we see considerable changes in respiration during speech compared to the baseline as well as to the listening episodes. One interesting difference we see is that for the first participant the chest and abdomen respiration differs more than for the second participation where both belt records seem to be similar.

These findings give an idea of possible respiratory changes during emotional states. Here, the most important observation for virtual agents is that the chest respiration should take precedence in respiration synthesis for virtual agents.

To evaluate the effects of music on respiration we also analysed the respiration data of all participants. Multiple statistical analyses were conducted, but due to space constraints, we only discuss the averaged respiration feature values for each emotion category.

Similar to our evaluation of valence and arousal levels, we compare respiration features of each emotion category to the baseline condition. We observe that among all the respiration features, there was only one statistically significant difference: between at least two of the mean chest respiration peak to peak distances during the happy–sad condition, $F(2, 20) = 39.75$, $p > 0.001$. A post hoc pairwise comparison of the least significant difference (LSD) type showed furthermore that there was a significantly higher peak to peak chest respiration distance for no music ($M = 4.41$, $SD = 0.79$) compared to sad music ($M = 3.70$, $SD = 0.61$), $p = 0.033$. This suggests that breathing cycles become shorter when listening to sad music, compared to no music listening. There was no significant difference between no music ($M = 4.41$, $SD = 0.79$) and happy music ($M = 3.70$, $SD = 0.64$), $p = 0.062$, nor between sad ($M = 3.70$, $SD = 0.61$) and happy music, $p = 0.982$.

Analysing the mean differences of respiration features, certain tendencies became visible. For example, sad music deepens the abdominal respiration as both the chest and abdominal respiration have high respiration frequencies. Happy music has a similar increasing effect on respiration frequency, but may induce a more shallow chest respiration pattern. Calm music shows a non-significant tendency to lower the abdominal respiration frequency and may increase chest respiration regularity. Annoying music shows an opposite pattern, where the abdominal respiration frequency is higher and the abdominal respiration becomes more irregular. The

results are summarised in Table 2. We also provide a summary for each songs effect in Appendix, Table 4.

## 5 CONCLUSIONS

We have introduced the Emotional Respiration Speech Dataset, consisting of audio recordings of 20 participants while they listened to self-selected songs that trigger intense emotions from the two emotion pairs of happy/sad or calm/annoying. The dataset furthermore is enriched by ground-truth breathing features collected via piezo-electric respiration sensors, as well as SAM questionnaires filled-in during the experiment for self-reported valence and arousal levels.

The unique setup of the dataset creates some specific challenges for emotion elicitation for speech, which can be considered as to be typical for emotion elicitation experiments. Even though the relation between music and emotion elicitation has established strongly in the literature, this dataset is the first to use music for emotional speech. As we base our analysis on self-reports at the end of each listening episode, we do not have a more nuanced labeling throughout the listening and speech episodes. We also need to further test if the music induced emotion is similar to emotions experienced and expressed during conversations. Lastly, we need to note that there is a potential bias that comes with self annotations. Despite these shortcomings, the analysis of the dataset still brought interesting insights and future directions.

We have observed that the self-reported valence levels were triggered by the chosen music for all four emotion categories as reported in the literature, whereas the changes in the arousal values were not significant. The amount of time participants were exposed to music might have played a role. The analysis of breathing patterns confirmed the literature that one has a relatively high respiration rate when listening to happy music.

Our qualitative observations of respiration patterns during speech and listening renders the differences between participants for all conditions, i.e. from the baseline to listening to emotional songs as well as speaking about them. Another important point here is the differences observed between chest and abdomen signals of respiration that can be similar or different according to individuals. An interesting future work could be to investigate if personality and mood has an affect on these features. Another line of investigation would be to test if the perception of personality would change with the addition of specific breathing features to virtual agents.

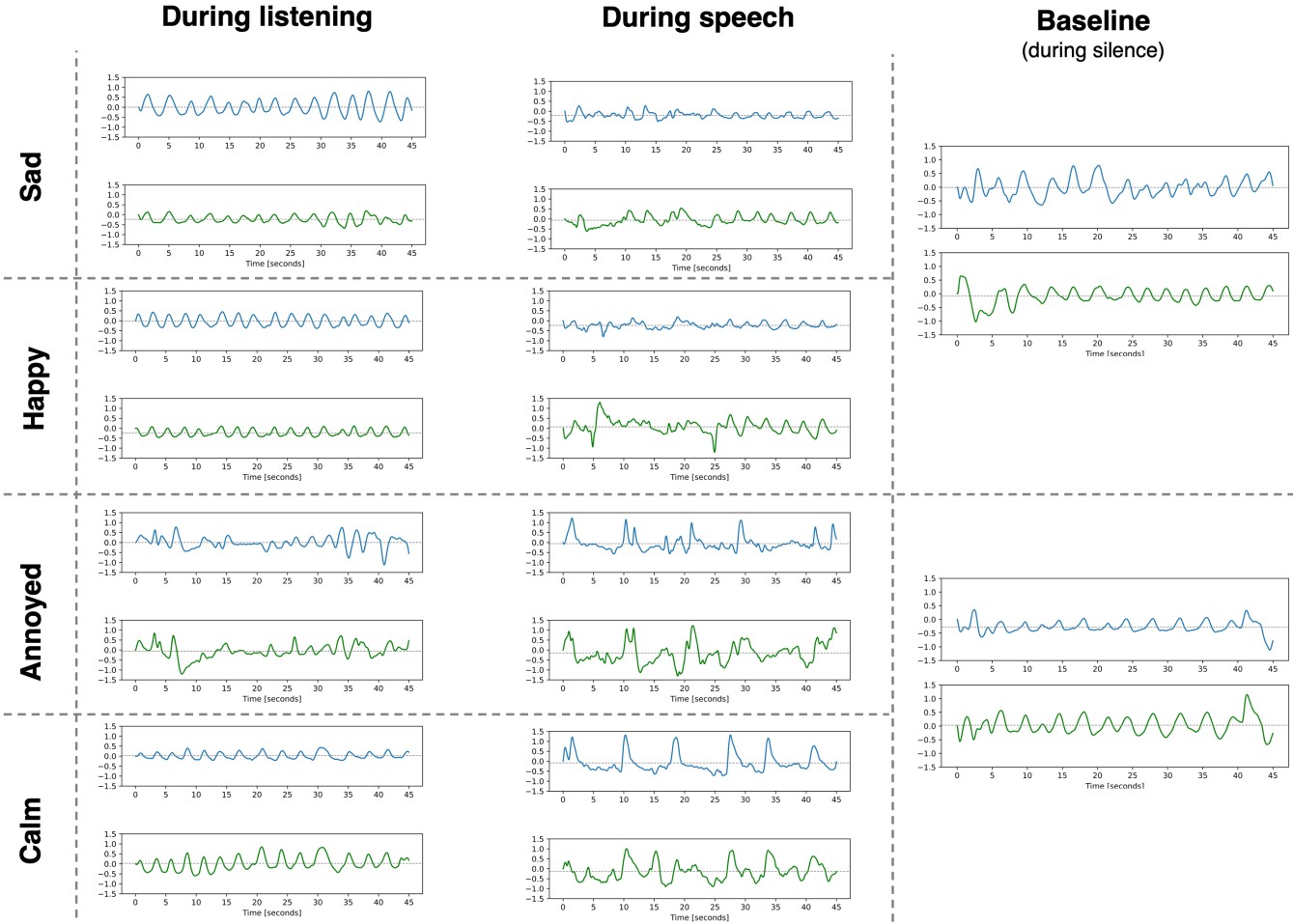

**Figure 5: Changes in respiration for two participants while listening to music (first column) and while speech (second column). The third column shows the baseline, respiration without music. Inhalation is represented by an increase in the signal, whereas exhalation is represented by a decline in the signal. The signal collected by the chest and abdominal respiratory belts are shown in blue and green, respectively. The first participant (upper graphs) is exposed to sad and happy songs whereas the second participant listened to annoying and calms songs.**

These results and harnessing the potential of Emotional Respiration Speech Dataset will be useful when modelling realistic embodied agents. To convey sadness or happiness, an agent should be programmed to have a higher respiration rate. Obviously, breathing patterns are just one aspect of creating the appearance of emotions, but they are strong and even a non-anthropomorphic design, such as a single robotic arm, becomes much more life-like via the addition of breathing motions [41]. Next to other non-verbal and verbal behaviours, respiration can also enhance the clarity of the conveyed emotions.

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

**Table 2: Mean respiration feature values during music listening for the two condition pairs and their baseline. RF = respiration frequency in Volt/seconds; PP = peak to peak distance in seconds; RD = respiration depth in Volt The asterisk (\*) indicates a significant effect of the emotion category on respiration values, compared to when listening to no music.**

| | Happy-sad condition (N=11) | | | Calm-annoying condition (N=9) | | |
|---|---|---|---|---|---|---|
| | No music | Happy | Sad | No music | Calm | Annoying |
| Chest RF (SD) | 16.19 (2.00) | 17.51 (±2.54) | 17.60 (±2.52) | 16.51 (±3.19) | 15.98 (±3.74) | 16.66 (±3.10) |
| Abdominal RF (SD) | 15.92 (±4.32) | 17.21 (±2.87) | 17.07 (±2.74) | 15.86 (±2.92) | 15.42 (±3.31) | 16.02 (±3.16) |
| Chest PP (SD) | 4.41 (±0.79) | 3.70 (±0.64) | **3.70 (±0.61)\*** | 4.20 (±0.76) | 4.32 (±1.19) | 3.97 (±0.91) |
| Abdominal PP (SD) | 1.27 (±1.76) | 1.48 (±2.53) | 1.19 (±1.07) | 2.03 (±1.67) | 2.15 (±1.81) | 2.04 (±1.59) |
| Chest RD (SD) | 0.64 (±0.29) | 0.54 (±0.29) | 0.62 (±0.30) | 0.65 (±0.38) | 0.65 (±0.21) | 0.62 (±0.13) |
| Abdominal RD (SD) | 0.54 (±0.23) | 0.52 (±0.24) | 0.71 (±0.42) | 0.56 (±0.19) | 0.61 (±0.15) | 0.62 (±0.20) |

general well being and heart rate variability. *Journal of the Indian Medical Association* 111, 10 (2013), 662–665.

[10] Celso M de Melo, Patrick Kenny, and Jonathan Gratch. 2010. Real-time expression of affect through respiration. *Computer animation and virtual worlds* 21, 3-4 (2010), 225–234.

[11] Ranit Gabriely, Ricardo Tarrasch, Maria Velicki, and Zehava Ovadia-Blechman. 2020. The influence of mindfulness meditation on inattention and physiological markers of stress on students with learning disabilities and/or attention deficit hyperactivity disorder. *Research in developmental disabilities* 100 (2020), 103630.

[12] Steven M. Gillespie, Ian J. Mitchell, Dawn Fisher, and Anthony R. Beech. 2012. Treating disturbed emotional regulation in sexual offenders: The potential applications of mindful self-regulation and controlled breathing techniques. *Aggression and Violent Behavior* 17 (7 2012), 333–343. Issue 4. https://doi.org/10.1016/J.AVB.2012.03.005

[13] Ikuo Homma and Yuri Masaoka. 2008. Breathing rhythms and emotions. *Experimental physiology* 93, 9 (2008), 1011–1021.

[14] Paweł Jemioło, Dawid Storman, Barbara Giżycka, and Antoni Ligęza. 2021. Emotion elicitation with stimuli datasets in automatic affect recognition studies–umbrella review. In *IFIP Conference on Human-Computer Interaction*. Springer, 248–269.

[15] Ravinder Jerath, Molly W Crawford, Vernon A Barnes, and Kyler Harden. 2015. Self-regulation of breathing as a primary treatment for anxiety. *Applied psychophysiology and biofeedback* 40, 2 (2015), 107–115.

[16] Mahin Kamalifard, Mahnaz Shahnazi, Manizheh Sayyah Melli, Shirin Allahverdizadeh, Shiva Toraby, and Atefeh Ghahvechi. 2012. The efficacy of massage therapy and breathing techniques on pain intensity and physiological responses to labor pain. *Journal of caring sciences* 1, 2 (2012), 73–78.

[17] Troels Aske Klausen, Ulrich Farhadi, Evgenios Vlachos, and Jonas Jørgensen. 2022. Signalling Emotions with a Breathing Soft Robot. In *2022 IEEE 5th International Conference on Soft Robotics (RoboSoft)*. IEEE, 194–200.

[18] Sylvia D. Kreibig. 2010. Autonomic nervous system activity in emotion: A Review. *Biological Psychology* 84, 3 (2010), 394–421. https://doi.org/10.1016/j.biopsycho.2010.03.010

[19] Helen Lavretsky and Jack L Feldman. 2021. Precision Medicine for Breath-Focused Mind-Body Therapies for Stress and Anxiety: Are We Ready Yet? *Global Advances in Health and Medicine* 10 (2021), 2164956120986129.

[20] Sadaf Lotfalian, Claire A Spears, and Laura M Juliano. 2020. The effects of mindfulness-based yogic breathing on craving, affect, and smoking behavior. *Psychology of Addictive Behaviors* 34, 2 (2020), 351.

[21] Yuri Masaoka and Ikuo Homma. 2001. The effect of anticipatory anxiety on breathing and metabolism in humans. *Respiration physiology* 128, 2 (2001), 171–177.

[22] Yuri Masaoka, Masahiko Izumizaki, and Ikuo Homma. 2014. Where is the rhythm generator for emotional breathing? *Progress in brain research* 209 (2014), 367–377.

[23] Yuri Masaoka, Haruko Sugiyama, Atsushi Katayama, Mitsuyoshi Kashiwagi, and Ikuo Homma. 2012. Slow breathing and emotions associated with odor-induced autobiographical memories. *Chemical senses* 37, 4 (2012), 379–388.

[24] Kevin D McCaul, Sheldon Solomon, and David S Holmes. 1979. Effects of paced respiration and expectations on physiological and psychological responses to threat. *Journal of Personality and Social Psychology* 37, 4 (1979), 564–571.

[25] James Nestor. 2020. *Breath: The new science of a lost art.* Penguin.

[26] Radosław Niewiadomski and Catherine Pelachaud. 2012. Towards multimodal expression of laughter. In *International Conference on Intelligent Virtual Agents*. Springer, 231–244.

[27] Radosław Niewiadomski, Jérôme Urbain, Catherine Pelachaud, and Thierry Dutoit. 2012. Finding out the audio and visual features that influence the perception of laughter intensity and differ in inhalation and exhalation phases. In *Proceedings of 4th International Workshop on Corpora for Research on Emotion, Sentiment & Social Signals, LREC*.

[28] David Novick, Mahdokht Afravi, and Adriana Camacho. 2018. PaolaChat: a virtual agent with naturalistic breathing. In *International Conference on Virtual, Augmented and Mixed Reality*. Springer, 351–360.

[29] Lauri Nummenmaa, Enrico Glerean, Riitta Hari, and Jari K. Hietanen. 2014. Bodily maps of emotions. *Proceedings of the National Academy of Sciences* 111, 2 (2014), 646–651. https://doi.org/10.1073/pnas.1321664111 arXiv:https://www.pnas.org/doi/pdf/10.1073/pnas.1321664111

[30] Pierre Philippot, Gaëtane Chapelle, and Sylvie Blairy. 2002. Respiratory feedback in the generation of emotion. *Cognition & Emotion* 16, 5 (2002), 605–627.

[31] Pierre Philippot, Gaëtane Chapelle, and Sylvie Blairy. 2002. Respiratory feedback in the generation of emotion. *Cognition & Emotion* 16, 5 (2002), 605–627.

[32] Tiago Ribeiro and Ana Paiva. 2012. The illusion of robotic life: principles and practices of animation for robots. In *Proceedings of the seventh annual ACM/IEEE international conference on Human-Robot Interaction*. 383–390.

[33] Amélie Rochet-Capellan and Susanne Fuchs. 2014. Take a breath and take the turn: how breathing meets turns in spontaneous dialogue. *Philosophical Transactions of the Royal Society B: Biological Sciences* 369, 1658 (2014), 20130399.

[34] James A Russell. 1980. A circumplex model of affect. *Journal of personality and social psychology* 39, 6 (1980), 1161.

[35] Guy Santibanez-H. and Susana Bloch. 1986. A qualitative analysis of emotional effector patterns and their feedback. *The Pavlovian Journal of Biological Science* 21, 3 (1986), 108–116. https://doi.org/10.1007/bf02699344

[36] Ameneh Shamekhi and Timothy Bickmore. 2015. Breathe with me: a virtual meditation coach. In *International Conference on Intelligent Virtual Agents*. Springer, 279–282.

[37] Ameneh Shamekhi and Timothy Bickmore. 2018. Breathe deep: A breath-sensitive interactive meditation coach. In *Proceedings of the 12th EAI International Conference on Pervasive Computing Technologies for Healthcare*. 108–117.

[38] HUR Siddiqui, Hf Shahzad, AA Saleem, AB Khan Khakwani, F Rustam, E Lee, I Ashraf, and S Dudley. 2021. Respiration Based Non-Invasive Approach for Emotion Recognition Using Impulse Radio Ultra Wide Band Radar and Machine Learning. *Sensors* 21 (2021), 8336.

[39] Rukshani Somarathna, Tomasz Bednarz, and Gelareh Mohammadi. 2022. Virtual Reality for Emotion Elicitation–A Review. *IEEE Transactions on Affective Computing* 13, 01 (2022), 1–21.

[40] DC Stanescu, BENOfT Nemery, CLAUDE Veriter, and CLAUDE Marechal. 1981. Pattern of breathing and ventilatory response to CO2 in subjects practicing hatha-yoga. *Journal of Applied Physiology* 51, 6 (1981), 1625–1629.

[41] Yunus Terzioğlu, Bilge Mutlu, and Erol Şahin. 2020. Designing Social Cues for Collaborative Robots: The RoIe of Gaze and Breathing in Human-Robot Collaboration. In *2020 15th ACM/IEEE International Conference on Human-Robot Interaction (HRI)*. IEEE, 343–357.

[42] Frank Thomas, Ollie Johnston, and Frank Thomas. 1995. *The illusion of life: Disney animation*. Hyperion New York.

[43] Helen Y. Weng, Jack L. Feldman, Lorenzo Leggio, Vitaly Napadow, Jeanie Park, and Cynthia J. Price. 2021. Interventions and Manipulations of Interoception. *Trends in Neurosciences* 44, 1 (Jan. 2021), 52–62. https://doi.org/10.1016/j.tins.2020.09.010

[44] Hilal Yuksel, Yasemin Cayir, Zahide Kosan, and Kenan Tastan. 2017. Effectiveness of breathing exercises during the second stage of labor on labor pain and duration: a randomized controlled trial. *Journal of integrative medicine* 15, 6 (2017), 456–461.

[45] Kun Zhou, Berrak Sisman, Rui Liu, and Haizhou Li. 2022. Emotional Voice Conversion: Theory, Databases and ESD. arXiv:2105.14762 [cs.CL]

## 6 APPENDIX

**Table 3: Mean valence and arousal values during separate song listening for the two condition pairs and their baseline. The asterisk (\*) indicates a significant difference in dimensional values between the different songs for each emotion category and no music (baseline).**

| Music emotion | Mean valence (SD) | F (p-value) | Mean arousal (SD) | F (p-value) |
|---|---|---|---|---|
| No music (N=11) | 7.00 (±0.89) | | 4.73 (±1.42) | |
| First happy song (N=11) | 7.73 (±0.91) | | 5.45 (±1.75) | |
| Second happy song (N=11) | 7.64 (±0.92) | | 6.18 (±1.54) | |
| First sad song (N=11) | **3.82 (±1.33)\*** | | 4.27 (±1.56) | |
| Second sad song (N=11) | **3.73 (±1.35)\*** | 45.7 (**<.001**) | 5.09 (±0.94) | 2.97 (.063) |
| No music (N=9) | 6.78 (±0.67) | | 4.89 (±1.05) | |
| First calm song (N=9) | 7.67 (±0.71) | | 4.78 (±2.33) | |
| Second calm song (N=9) | 7.56 (±1.13) | | 3.78 (±2.11) | |
| First annoying song (N=9) | **4.33 (±1.80)\*** | | 6.22 (±1.30) | |
| Second annoying song (N=9) | **4.22 (±1.72)\*** | 28.18 (**<.001**) | 6.11 (±1.62) | 3.34 (.059) |

**Table 4: Mean respiration feature values during separate songs, compared to the other emotion pair and no music listening. There were no significant differences. RF = respiration frequency in Volt/seconds; PP = peak to peak distance in seconds; RD = respiration depth in Volt.**

| | Emotion category of separate song | | | | | |
|---|---|---|---|---|---|---|
| N = 11 | No music | First happy song | Second happy song | First sad song | Second sad song | F-value (p) |
| Chest RF (SD) | 16.19 (±2.00) | 17.11 (±2.82) | 17.90 (±2.97) | 17.84 (±2.61) | 17.36 (±2.87) | 1.25 (.305) |
| Abdominal RF (SD) | 15.92 (±4.32) | 17.48 (±2.83) | 16.95 (±5.25) | 17.90 (±2.47) | 16.25 (±4.06) | 0.15 (.709) |
| Chest PP (SD) | 4.41 (±0.79) | 3.82 (±0.83) | 3.57 (±0.62) | 3.68 (±0.66) | 3.72 (±0.73) | 3.03 (.059) |
| Abdominal PP (SD) | 1.27 (1.76) | 0.96 (±1.05) | 1.99 (±4.16) | 1.11 (±1.07) | 1.28 (±1.08) | 0.79 (.400) |
| Chest respiration depth in volt (SD) | 0.64 (±0.29) | 0.54 (±0.29) | 0.50 (±0.29) | 0.59 (±0.31) | 0.66 (±0.31) | 0.72 (.492) |
| Abdominal respiration depth (SD) | 0.54 (±0.23) | 0.54 (±0.36) | 0.51 (±0.19) | 0.65 (±0.40) | 0.76 (±0.48) | 1.54 (.209) |
| N = 9 | No music | First calm song | Second calm song | First annoying song | Second annoying song | F-value (p) |
| Chest RF (SD) | 16.51 (±3.19) | 16.66 (±3.63) | 15.30 (±4.40) | 16.60 (±3.29) | 16.71 ±3.07) | 0.81 (.466) |
| Abdominal RF (SD) | 15.86 (±2.92) | 15.26 (±3.48) | 15.58 (±3.18) | 16.02 (±3.62) | 16.02 (±2.80) | 0.26 (.707) |
| Chest PP (SD) | 4.20 (±0.76) | 4.09 (±1.05) | 4.54 (±1.55) | 4.01 (±0.96) | 3.93 (±0.89) | 1.25 (.313) |
| Abdominal PP (SD) | 2.03 (±1.67) | 2.16 (±1.83) | 2.14 (±1.80) | 2.05 (±1.47) | 2.04 (±1.73) | 0.30 (.744) |
| Chest respiration depth in volt (SD) | 0.65 (±0.38) | 0.61 (±0.22) | 0.68 (±0.22) | 0.57 (±0.15) | 0.67 (±0.22) | 0.37 (.647) |
| Abdominal respiration depth (SD) | 0.56 (±0.19) | 0.65 (±0.16) | 0.57 (±0.22) | 0.74 (±0.29) | 0.52 (±0.24) | 2.72 (.107) |