# OpenReview forum: "Emotional Respiration Speech Dataset"
_ACM.org/ICMI/2022/Workshop/GENEA — GENEA Challenge & Workshop 2022 Workshopproceeding_

### Official Review · Reviewer_GCab · 2022-08-09
**Review of Emotional Respiration Speech Dataset**

**Rating:** 4
**Confidence:** 4

**Review:**

This paper describes a new dataset that was elicited by having participants listen to self-chosen music while being monitored, and using self-reported emotion data. The purpose of this is to create a dataset of affective respiration that could be used to improve social non-verbal communication between humans and VAs or robots.

I come in skeptical that “music-induced emotion” is the same as conversational emotion, so it seems like the task and dataset doesn’t relate to the intended application.

But interesting core concept of paralinguistic cues.

Enjoyed the introduction; good overview of problem space and appropriate background and context for the challenge. However, “Neither of these types of emotional datasets replicate the conversational or spontaneous speech, and lack in ecological validity” – I feel that then proposing listening to music to elicit emotion doesn’t address these challenges very well.

Especially appreciated the related work section. As a NVB researcher not very familiar with paralinguistic cues it provides a strong overview of the state of the field, as well as the impetus behind the behavior (ANS and emotion regulation).

In section 2.3, I would note that forced-choice emotion evaluation is notoriously fickle – being able to distinguish beteween 3 or 4 emotions is not the same as using those cues in real life, especially with such a small-dimensional expression such as respiration. For example, the authors themselves mention both digust and pleasure elicit slower breathing, whereas anxiety and anger both elicit shallow rapid breathing. I find these studies dubious, and the authors put a lot of emphasis on them. The main takeaway – “they do make clear that humans use some sort of induction of one’s emotions be observing the respiration” seems like the big takeaway and should be emphasized more. Especially since the evaluation of the dataset does not depend on this forced choice emotion recognition.

On that note – authors use “emotion detection” quite often but is it genuine detection? Detection would be unprompted emotion recognition, whereas the tasks the authors often describe seems to me to be forced-choice emotion recognition.

Why not use existing speech datasets annotated with emotional information and using respiration patterns detected from microphones/video?

3 Experiment
Having people select their own music reinforces individual differences in emotion interpretation and understanding; not only will the music differ but what “happy” means to each participant islikely extremely different, especially in the context of music.
University-student study is not representative of population at large; also absolutely necessary to say where the students are from as expression of emotion varies hugely between e.g. Europe (even countries within) and Japan.

4 Results
Asking people to self-report their emotions after listening to the songs they said would elicit those emotions is not a strong design; they are severely biased to believe they are in the emotional state they said they’d be in because they chose the song.

Overall while the concept for this database is strong and the authors provide a good case for it in VA research, the elicitation and evaluation methods are not suitable for the task. While it is, again, an interesting concept that just needs some work, it also does not seem relevant to the GENEA gesture challenge.

---

### Official Review · Reviewer_dy6G · 2022-08-16
**see below**

**Rating:** 8
**Confidence:** 5

**Review:**

The paper describes a new database to study breathing while speaking. The paper presents the design of the corpus, a description of the corpus, and a preliminary analysis of the data. Overall, this is an interesting, well-written paper. It fits the scope of the workshop, since this data can be used to synthesize virtual agents expressing realistic breathing patterns.

The paper discusses several results reported on previous studies about the relation between “emotional classes” and respiration, some of them reporting conflicting results. My view on this issue is that several realizations of an “emotion” can be quite diverse. For example, cold anger is quite different from hot anger. Similar observations can be made for happiness, when several varied behaviors can be clustered under the term “happiness.” In this context, some strong manifestations of an emotion can produce a breathing change, but more mild versions the emotion may not.

Can we change emotion so quickly from one emotion to another by just listening 90min music?

There is also potential bias on the self annotations. If you ask me what are the songs that make me feel “happy,” and then you play that song to me and ask me how I feel, chances are that I am going to say happy. I understand that you are annotating arousal and valence, but this is still a problem.

Another concern is the lack of temporal resolution on the annotations. They provide a single ranting for the entire time that they spoke. Emotions fluctuate when we interact with other. This variations are not captures by the labels.

The size of the corpus is quite small.

Is the study of personality suggested in the conclusion even possible with this corpus? did you collect personality information from your speakers? You only have 20 subjects, which is small for studying the role of personality.



Minor comments:

“In an interesting 73 study, Terzioğlu et al. have shown” add reference.

“Modelling realistic emotionality” What do you mean?

“user will also contain significant importation about their emotional state” importation -> information

“if the addition of realistic breathing adds to more rapport and natural perception of a virtual agent” (I believe this sentence reads better if “to” is removed)

“the separate song values are not displayed in this paper” can you add this in the appendix?

“but due to brevity reasons” ?  space constraints?

“Sad music probably has the same effect, but this needs further research.” If your results do not show this, my recommendation is do not speculate.

---

### Decision · Program_Chairs · 2022-08-20

**Decision:**

Accept (Workshop proceeding)

**Comment:**

This paper received mixed reviews. Both reviewers mentioned that the topic is interesting and that the paper well described the problem and previous studies. The main weakness of this paper is the data collection settings that were not fully validated and controlled. For instance, it is not obvious that music-induced emotion represents conversational emotion; there are biases on participants, music selection, and annotations, and these biases matter a lot in the small size of the dataset.

Despite the limitations, this paper could stimulate further research on emotional respiration so we, the program chairs, would like to accept this paper as a GENEA workshop paper.

Please read the reviews carefully and revise the paper. In particular, we recommend adding a discussion for the limitation of the dataset so that readers be aware of the them.